# Intelligent Management of Chemical Warehouses with RFID Systems

**DOI:** 10.3390/s20010123

**Published:** 2019-12-24

**Authors:** Jumin Zhao, Fangfang Xue, Deng-ao Li

**Affiliations:** 1College of Information and Computer, Taiyuan University of Technology, Taiyuan 030024, China; xuefangfang67@163.com; 2Technology Research Center of Spatial Information Network Engineering of Shanxi, Taiyuan 030024, China; lidengao@tyut.edu.cn; 3College of Data Science, Taiyuan University of Technology, Taiyuan 030024, China

**Keywords:** RFID, detect, remaining amount, localization

## Abstract

At present, most chemical warehouses rely on human management, which is a time-consuming and laborious process. Therefore, it is very meaningful to use radio frequency identification (RFID) systems for the intelligent management of chemicals. Detecting the remaining amount of chemicals is an important process in the management of a chemical warehouse. It helps managers find the chemicals that are going to run out and replenish them in time. However, in a traditional chemical warehouse, managers usually inspect each chemical on the shelf in turn manually, which is a waste of time and labor. Although some solutions using RFID technology have been proposed, they are expensive and difficult to deploy in a real environment. In order to solve this problem, we propose an intelligent system called the RF-Detector in this paper, which combines robotics and RFID technology. An RFID reader and an antenna are installed on the robot, which achieves automatic scanning of the chemicals. The RF-Detector can achieve two functions: One function is to detect the remaining amount of chemicals using the changes in received signal strength indication (RSSI) and read rate, and the other is to locate chemicals using the phase curve, so that managers can quickly find the chemicals with an insufficient amount remaining. In this paper we implement the RF-Detector and evaluate its performance. The experimental results show that the RF-Detector achieves about 93% detection accuracy and 92% positioning accuracy for chemicals.

## 1. Introduction

The management of chemical inventory in a traditional chemical warehouse is difficult. Thousands of chemicals are displayed in any chemical warehouse. This is demonstrated in Figure 1, which is just a corner of the chemical warehouse. Managers need to check the remaining amount of chemicals every day and replenish them in time. This management method requires large manpower, and may also result in statistical errors due to human error. Moreover, since chemicals may be placed in another location after use, it may cause difficulties in subsequent uses. Hence, the management and positioning of chemicals is a problem that must be solved.

In recent years, radio frequency identification (RFID) systems have been widely used in logistics tracking, warehouse management, mobile healthcare and other fields, and employing RFID to manage warehouse inventory has attracted increasingly attention [1,2,3]. For example, it has been widely used in supply chain management and in enabling automatic identification and tracking of goods [4]. AuRoSS [5] implements the automatic scanning of bookshelves by combining RFID and robot technology. OTrack [6] uses mobile RFID systems to track luggage on airport conveyors. But the application of RFID in a chemical warehouse is still immature. Some chemistry laboratories use handheld readers or deploy smart chemical management devices to reduce the burden on managers [7,8,9]. However, both methods have significant shortcomings. For the handheld reader, managers still need to manually move the reader to scan each container, which is time consuming and laborious. For the smart devices, the process is costly and requires re-deployment of warehouse hardware, which is not easy to achieve.

In this paper, we design a system called RF-Detector, which can detect the remaining amount of chemicals, find out which chemicals are going to run out and find their position, so that managers can supplement them in time. We attach an RFID tag, which contains information about the chemical, to each chemical container at a height of 1 cm from the bottom of the bottle. The RF-Detector consists of a robot and the RFID. We install the RFID reader and antenna on the robot, and it scans the tags as the robot moves [10,11]. The RF-Detector can realize automatic scanning and detection, reducing manpower consumption greatly, and it is of low cost and easy to deploy, which makes it applicable in practice. In this paper, we borrow the robot technique to achieve the shelf scanning automatically and efficiently, so we pay more attention to the related functions of RFID instead of focusing on the principle of the robot. Two issues are solved by RF-Detector in the chemical warehouse.

The first is how to detect the remaining amount of chemicals in the container. We know that some containers are transparent and some are opaque, and the chemical in the bottle may be solid or liquid, so these factors will hinder our testing. In order to check the remaining amount of each chemical, managers need to check all containers one by one. Many containers are opaque, so they need to unscrew the cap to check, and there is a large variety of chemicals, so this work is time consuming and labor intensive. By careful observation, we find two useful metrics, received signal strength indication (RSSI) and the read rate. We combine two indicators together and take them as the input of the support vector machine (SVM) to distinguish the chemicals which are going to run out and which are sufficient.

The second is how to locate chemicals that are going to run out. Location information should indicate which shelf, which tier, and which order they are in, allowing managers to find the almost exhausted chemicals easily and replenish them in time. There are two problems in the positioning process. The first is that when scanning a tag on a chemical rack, the tag on the upper tier and the lower tier may participate in the response, which will have a certain impact on the tag positioning. The second problem is that due to the types of chemicals in the container and the remaining amount of chemical being different, the phase value of the tag will be affected, which may interfere with the positioning of the tag. We use the phase value of the tag to locate the tag: The phase value of each tag changes over time to form a phase curve, so by fitting this curve we can determine the location of the tag according to the parameters of the curve.

We install a RFID reader Impinj R420 and an antenna on the robot. The robot can walk along the shelves through magnetic navigation and provide conditions for continuous and stable scanning of the RFID reader. The antenna is attached to the arm of the robot and can be moved up and down to scan each layer of the shelf. The robot records the currently scanned shelf number and the layer number at any time during the movement.

We define the notions that will be used below:

**Received signal strength indication (RSSI):** A measurement of the strength of a radio signal being received. The radio signal will lose energy during transmission. The more the loss, the lower the signal strength of the signal returned by the tag.

**Read rate:** The read rate is the number of times a single tag can be read within a given period divided by the maximum number of reads of the single tag. According to the type of reader and tag, the frequency and the power of the transmitted signal of the antenna, we can query the theoretical reads of the single tag within a given period in the manual, which can be regarded as the maximum number of reads of the single tag. The actual number of times a single tag that can be read within a given period is affected by the chemicals in the bottle, is lower than the theoretical number of reads and its value can be reported by the reader. The ratio of the two is the read rate of the tag.

**Detection accuracy:** The detection accuracy is the number of correct judgments regarding whether the remaining amount of chemicals is sufficient or insufficient, divided by the total number of experiments.

**Positioning accuracy:** The positioning accuracy is the number of successful positioning to the corresponding position divided by the total number of experiments.

We have three contributions in this paper. First, we combine RFID technology and robot technology and propose the RF-Detector system, which can scan chemicals automatically. Second, we combine the RSSI and the read rate to achieve the detection of the remaining amount of chemicals in the container, achieving a detection accuracy of 93%. Third, we are able to locate chemicals with a positioning accuracy of 92%, which helps managers find the right chemicals quickly and accurately.

We next present the system architecture of RF-Detector in Section 2. Section 3 introduces the method of detecting the remaining amount of chemicals. Section 4 introduces the localization of chemicals. Section 5 describes the experiments. In Section 6, we describe the results and discussion. We present the related work in Section 7 and conclude in Section 8.

## 2. System Architecture

Our system uses a robot, an ImpinJ R420 reader, an Alien ALR-8696-C directional antenna, and some Alien ALR-9610 passive tags. To perceive all chemicals, each container is labeled with an RFID tag that records information about chemicals. The RF-Detector uses a magnetic navigation system, and the robot can detect the magnetic signals of the tracks laid on the ground in advance to achieve automatic scanning guidance. The robot is a tool to replace people, carrying RFID readers for mobile scanning. It uses the existing robotic system for scanning shelves, and can achieve three functions: The auto-navigation module enables automatic navigation so that it can carry a RFID scanner to auto-scan according to our pre-arranged route; the automatic charging module provides enough power for the robot’s work, ensuring that the robot can achieve continuous navigation; the robot has an automatic storage function that records the current scanned shelf number and the height of the robot arm, providing clues to the positioning of the tag. The directional antenna is a type of antenna that emits and receives particularly strong electromagnetic waves in one specific direction, and extremely small waves in other directions. The radiation pattern of the antenna is shown in the Figure 2. The RF-Detector uses the ultra-high frequency (UHF) sent by the antenna to scan chemicals, and the frequency range is 860–960 MHz. We don’t use a high frequency (HF) of 13.56 MHz, which is generally used in RFID applications with high security requirements such as identity, library management, and product management, because its communication range is short. UHF is used as it allows the system to have a high read rate, which ensures the reliable reading of tags in the mobile environment.

The following describes the working process of the RF-Detector. The reader moves to the starting point of the first chemical shelf. Next, the RFID system is activated and starts scanning the chemicals on the shelf. While scanning each shelf, the antenna is raised to the corresponding position to scan every layer. When the scan of one shelf is completed, the reader is moved to the next shelf and continues scanning. The above process is repeated until all chemicals have been scanned. The scanned data is used to realize the detection of the remaining amount and the positioning of chemicals.

The reader in the hardware system communicates with the RFID tags attached to the chemical bottles through electromagnetic wave transmission, and reads the tag number, RSSI value, phase value and other information. The communication between the hardware system and the host computer management software is achieved through the LLRP protocol provided by the reader. The host computer management software receives the tag information transmitted by the hardware part, uses the chemical detection algorithm to get the remaining amount information, and uses the positioning algorithm to get the chemical location information. The host computer management software communicates with the database using the HTTP protocol, and the management software uploads the collected tag information, location information, and remaining amount information of chemicals to the database, so that lab managers can obtain relevant data collected by the RF-Detector from the database.

## 3. Detecting the Remaining Amount of Chemicals

In this section, we focus on how to test for chemicals that are almost exhausted. If we only rely on manual inspection of each bottle of chemicals, this task is laborious. There are many ways to detect the remaining amount of liquid or solid. For example, some use a gravity sensor, as when the remaining amount of chemicals changes, the parameters of the corresponding gravity sensor also change. When the remaining amount of chemicals is below a predetermined threshold, we need to supplement it. We can also use the photoelectric sensor to clamp an infrared sensor on the container. When the chemical in the container is running out, its liquid level will be lower than the level where the infrared sensor is located, which triggers the circuit and reminds the management personnel. But neither of these methods are feasible. Both of these methods require a set of devices for each container, which is costly and complicated, and not suitable for the storage management of a large number of chemicals. Moreover, in this scenario, detection with video cameras is not suitable, as because some chemicals are invisible, they add a lot of noise to the image in low light conditions. Moreover, many chemicals are stored in opaque chemical containers, which cannot be detected by the video camera. In view of the above reasons, we propose a method for detecting the remaining amount of chemicals using the RFID system. The specific process is given below.

### 3.1. Change in RSSI

Although the RF signal used in RFID technology can penetrate paper, wood, plastic and other materials, it is still affected by many factors, such as solids, liquids and other materials, as they will absorb some (Radio Frequency) RF signals. Therefore, when the RF signal passes through some objects, the RF signal received by the reader will change significantly. Based on this feature, we can use the RSSI indicator to detect the remaining amount of chemicals in the container. RSSI is a power indicator of the received radio signal. The received radio signal power can be expressed as:(1)$$P_{r}=\gamma P_{t}G_{r}^{2}G_{t}^{2}{\left(\frac{\lambda }{4\pi d}\right)}^{4},$$
where $P_{r}$ is the power of the received backscattered signal, $P_{t}$ is the transmitting power of the reader, γ is the loss in the backscatter transmission, the distance between the reader antenna and the tag antenna is *d*, $G_{r}$ is the gains of the reader antenna and $G_{t}$ is the gains of the tag antenna. Thus, RSSI can be defined as:(2)$$RSSI=10log\left(\frac{P_{t}}{1mw}\gamma G_{r}^{2}G_{t}^{2}{\left(\frac{\lambda }{4\pi d}\right)}^{4}\right).$$

We try to use the change in RSSI value to determine the amount of chemicals in the container. We prepare six identical glass bottles with a height of 18.5 cm and a diameter of 8.5 cm, which are filled with NaCl solution, HCl solution, NaOH solution, KOH powder, KCl powder and Mg powder respectively. The RFID tag is attached to the outside of the container at a height of 1 cm from the bottom. When the horizontal planes of the six chemicals are 0 cm, 3 cm, 6 cm, 9 cm, 12 cm, 15 cm and 18 cm, we measure the RSSI of the tag corresponding to every chemical, observing the change in RSSI as the liquid level rises. The measurement result is shown in Figure 3.

We find that the changes in RSSI of these six chemicals are different when the volume changes. As the liquid level rises, the RSSI of each chemical decreases, but the slope of the RSSI value of each chemical is different. The RSSI value of the solid decreases faster as the liquid level rises, and the RSSI value of the liquid is reduced relatively slowly.

### 3.2. Change in Read Rate

The read rate of the RFID tag is an important parameter, because as the chemicals in the container absorb the RF signal, its dielectric constant is very high at higher frequencies. For example, water’s dielectric constant can reach 70 at a frequency of 1 GHz. It will inevitably reduce the antenna gain of the tag and cause the tag antenna to be detuned, ultimately affecting the read rate of the tag [12].

We try to use the change in read rate to determine the amount of chemicals in the container. We repeat the experiment described earlier, when the horizontal planes of the six chemicals are 0 cm, 1 cm, 2 cm, 3 cm, 4 cm, 5 cm, 6 cm, 7 cm, 8 cm and 9 cm, and we measure the read rate of the tag corresponding to every chemical, observing the change in read rate as the liquid level rises. The measurement result is shown in Figure 4. We find that as the level of chemicals in the container rises, the read rate of the tag decreases, and when the liquid level changes from 1 cm to 2 cm, that is, when the liquid level exceeds the tag position, the reading rate drops sharply. When the actual read rate is greater than 90% of the maximum read rate, it can be determined that the level of the chemical is lower than the level of the tag attached to the outside of the container, indicating that the remaining amount of the chemical is insufficient.

### 3.3. Combining Two Indicators

In the above, we have discussed the effect of the remaining amount of chemicals on the RSSI of the tag and the frequency of reading. Compared to using only a single indicator, we can improve the accuracy of detection by combining the two indicators. We use the ID number of the tag, the corresponding RSSI, and the read rate as a four-element array ID, *t*, RSSI, *n* at a certain moment. We put two indicators together to form a feature vector <RSSI, *n*>, thereby reducing our problem to a binary classification problem. We take this feature vector as the input of SVM, using this classification algorithm to identify the chemicals that are going to run out or be sufficient. Because the method uses two indicators in combination, the accuracy of the detection will be higher. In the following, we will conduct a comprehensive evaluation of this method.

## 4. Localization of Chemicals

When the system detects a chemical that is about to run out, we are required to locate the chemicals to replenish them. We need to know where the chemicals are: which shelf, which level, and which order. Next we will discuss how to get these three position indicators.

### 4.1. Theoretical Analysis

The phase θ reflects the offset degree between the transmitted signal and the received signal by the reader antenna, ranging from 0 to 2π. We assume that the distance between the reader antenna and the tag is *d*. Due to the backscatter communication of the RFID system, the reader antenna sends radio frequency signals, and the tag returns the carrier signal carrying its own information to the reader antenna to form a round-trip, where the distance of the RF signal transmission during a communication is 2*d*. In addition to the offset degree caused by the distance, there are also some unavoidable effects caused by the transmission circuit of the reader, the reflection coefficient of the tag, the receiver circuit of the reader etc., which are indicated as $\theta_{TX}$,$\theta_{RX}$, and $\theta_{TAG}$; these effects will add an additional phase offset. So the phase measurement θ by the reader can be expressed as:(3)$$\left\{\begin{array}{c} \theta =\left(2\pi \times \frac{2d}{\lambda }+\mu \right)mod2\pi \\ \mu =\theta_{TX}+\theta_{RX}+\theta_{TAG}, \end{array}\right.$$
where λ is the wavelength, and μ indicates system noise.

The Impinj R420 reader can measure and report the phase value θ. As the robot moves along the shelf, the reader continually scans the tags. Each time the tag is scanned by the reader, a phase value is obtained, so as the reader scans continuously, it obtains a series of phase values of the tag. During the movement of the reader antenna along the chemical shelf, the distance d is first decreased and then increased for a certain tag on the shelf, so that the phase value of the tag is first decreased and then increased. However, since the phase value of the tag changes at [0, 2π), when the phase value decreases to 0, it will jump to 2π at the next moment and then begin to decrease; when the phase value increases to 2π, the next moment will jump to 0 and start to increase. The phase value periodically repeats between 0 and 2π, as shown in Figure 5. However, the sudden jump of the phase curve will affect our positioning. We need to turn it into a continuous curve, so that we can fit the phase curve to achieve higher precision positioning.

### 4.2. Processing of the Phase Curve

Sudden jumps between the adjacent phase curves causes discontinuity. Therefore, before positioning, we need to process the transition between the curves, that is, remove the periodicity of the phase profile. We can then perform period compensation for the jump of the phase curve. Firstly, we acquire all phase values by time; then, we compare the adjacent stages of the curve in turn, calling these two adjacent phases $\theta_{i}$ and $\theta_{i+1}$. If $\theta_{i+1}-\theta_{i}\approx 2\pi$, then it shows that there is a jump from near 0 to near 2π between these two adjacent phases. Therefore, all follow-up phases are $\theta_{j}$ (j>i) minus 2π. In contrast, if $\theta_{i}-\theta_{i+1}\approx 2\pi$, then it shows that there is a jump from near 2π to near 0 between these two adjacent phases. Therefore, all follow-up phases are $\theta_{j}$ (j>i) plus 2π. This process is repeated over the entire phase profile we measured until the phase values of the last two adjacent moments are compared.

Take Figure 6 as an example, from the first curve to the second, there is a jump from 0 to 2π, then for all the curves except the first minus 2π, so the second curve will subtract 2π because one 0–2π jump occurs in front, the third curve will subtract 2 × 2π because two 0–2π jumps occur in front, and the fourth curve will subtract 3 × 2π because three 0–2π jumps occur in front. After that, there will be jumps from 2π to 0, and every time it appears, all following phases plus 2π, and finally the super V region shown by the red curve in Figure 6 will be obtained. After the period compensation, all the phase curves are connected into a line, thus we eliminate the negative impact of the discontinuity of the phase curve on our positioning. Finally, after phase compensation, we get a super phase profile, as shown in Figure 6.

### 4.3. Localization of Chemicals by Curve-Fitting

In this part, we mainly introduce how we locate the chemical through the phase profile of the RFID tag. After the period compensation, the formula of the phase output should be re-written:(4)$$\left\{\begin{array}{c} \theta =2\pi \times \frac{2d}{\lambda }+\mu \\ \mu =\theta_{TX}+\theta_{RX}+\theta_{{}_{TAG}} \end{array}\right..$$

Through formula conversions, the super phase profile is theoretically simulated as a hyperbola that is subjected to a hyperbolic curve fitting [13], and finally the tag’s position is derived from the hyperbolic coefficient. The specific theoretical derivation process is as follows:

When the reader scans for chemicals, we set the starting point of each row of shelves as the origin of the coordinates, and establish a coordinate system in the plane parallel to the shelf. The moving speed of the antenna along the X axis is ν. Assuming that the height between the shelf layer and the layer is *h*, each time the antenna raises one level along the y axis, the height of the antenna is increased by *h*, and the robot can record the number *n* of layers raised by the antenna. As shown in Figure 7, if the coordinate of a certain tag is (a,b), the position of the antenna at time t is $\left(vt,nh\right)$, so the distance *d* between the tag and the antenna at time *t* is:(5)$$d=\sqrt{{\left(a-vt\right)}^{{}^{2}}+{\left(b-nh\right)}^{2}}.$$

Hence, the above formula can be changed as follows:(6)$$\theta =\frac{4\pi }{\lambda }\sqrt{{\left(a-vt\right)}^{{}^{2}}+{\left(b-nh\right)}^{2}}+\mu .$$

In Equation (Equation 6), *a* and *b* are of concern to us, λ,μ,v,n and *h* are known, *t* and θ are variables. The above formula can be changed as follows:(7)$$\frac{{\left(\theta -\mu \right)}^{2}}{{\left(\frac{4\pi }{\lambda }\right)}^{2}{\left(b-nh\right)}^{2}}-\frac{{\left(t-\frac{a}{v}\right)}^{2}}{\frac{{\left(b-nh\right)}^{2}}{{}^{v^{2}}}}=1.$$

Equation (Equation 7) is a form of hyperbolic function, which is fitted by a hyperbolic model, and finally the values of *a* and *b* are obtained to determine the position of the tag. The larger *a* is, the farther the chemical is from the starting position of the shelf; the larger *b* is, the higher the layer in which the chemical is on is located. According to the *a* and *b*, the location of the chemical can be obtained, and combined with the shelf number recorded during the robot navigation process, managers can quickly find the chemicals they are looking for.

## 5. Experiments

In the above, we have proposed that the reading rate of the tag is affected by the chemical in the bottle, so we observe the changes in the tag read rate when the chemicals are sufficient and insufficient through experiments. We fill the six bottles with the chemicals mentioned in Section 3.1, so that the liquid level is higher than 1 cm, which means that the remaining amount is sufficient, and test the reading rate of the tags. Then, we pour some liquid from the bottles to make the liquid level less than 1 cm, so that the remaining amount is insufficient, and we measure the read rate of the tags again. We repeat the experiments 20 times. Figure 8 and Figure 9 shows the reading rate of the tags corresponding to each chemical when the amount of chemical in the bottle is sufficient or insufficient. According to the experiment, when the remaining amount of chemicals is insufficient, the reading rate of the corresponding tag will increase, and the reading rate will be higher than 90%.

We simulate the scene of a simple chemical warehouse in the lab. We put 60 bottles on two shelves, where there are three floors in a shelf, and we put 10 bottles on each floor, filling these bottles with various chemicals. The RFID tag is attached to the outside of the bottle at a height of 1 cm from the bottom, and the bottles are placed in strict accordance with the rules of the chemical laboratory, with all RFID tags placed forward. Bottles #1 to #10 are placed on the first floor, #11 to #20 are placed on the second floor, and #21 to #30 are placed on the third floor. Bottles #31 to #60 bottles are placed on the other shelf in the same way.

Take the first floor of the first shelf as an example, the liquid level of the chemical in the #1, #2, #3, #4, #7 bottles is less than 1 cm, and the remaining amount of chemicals in other bottles is sufficient. First, we detect the remaining amount of chemicals. The robot moves the RFID reader and antenna to the back of the shelf for scanning. During each scan, the robot moves linearly along the X-axis with a moving speed of 0.2 m/s. We performed 20 experiments, recording the RSSI value and read rate of the tags for each scan. We analyzed the results of the first layer of the first shelf. Figure 10 shows the average RSSI for each tag of the first layer during 20 scans. Figure 11 shows the read rate for each scan. We found that in most cases, the experimental results were the same as we expected. The RSSI and read rate of the tag for the chemical with insufficient margins was higher than the tag with sufficient margin. However, there are some special cases, such as #4 and #8, where the read rate of the three scans did not meet our theoretical analysis. There are two reasons for these deviations: One is the multipath effect in our experimental environment, which made our measured RSSI less accurate; another reason is the delay caused by the C1G2 protocol, which uses the frame slot ALOHA algorithm. During the reading process, each tag randomly selects the time, and only when the tag selects a single time slot can it be successfully read; otherwise, it will reselect the time slot in the next frame. So some tags may have been successfully identified in the last few frames, which will have had a certain degree of impact on the read rate.

By using the RSSI and read rate of the tag, we can detect the chemicals that are going to run out through the SVM classifier. The next task is to locate the chemicals that are going to run out. In order to reduce the influence of the medium in the bottle on the positioning accuracy, we scan the tag directly from the front side of the rack without passing through the chemicals. The reader scans all the tags on the shelf again to get the phase change of every tag, extracting the phase value of the tags attached to the chemicals which are about to run out. Figure 12 shows the measured phase profile, and we can get the position of the tags by curve fitting.

## 6. Results and Discussion

In this section, we discuss two issues: One is the detection accuracy of the remaining amount of chemicals, and the other is the positioning accuracy of the chemicals. In the experimental scenario which we deployed, we repeat 20 trials and change the remaining amount and location of the chemical after every five experiments to maximize the mimicking of the chemical warehouse scenario.

We evaluate the accuracy of chemical residual detection first. During each experiment, we record the RSSI and read rate of the tags. Based on these two indicators, we finally construct a classification model that can determine whether the remaining amount of chemicals is sufficient. The accuracy of this method for detecting the remaining amount of chemicals that we achieved was 93%. Although this accuracy is not perfect, it can save the effort of managers from checking the chemicals in turn. Therefore, our work has a certain practical application value.

In the process of positioning chemicals, we need to know the positioning parameters shelf #, tier #, and order. The shelf # is recorded by the internal memory of the robot, and we pay attention to whether the tier # and the order are correct, that is, whether the values of a and b obtained after curve fitting are accurate.

We call the two shelves A and B, and the first layer of A is A1. To illustrate the positioning accuracy of this system, we use a confusion matrix to illustrate. For the positioning result of each tag, if the predicted shelf number and floor number is not the same as the ground truth, we believe it to be wrong positioning. The confusion matrix in Figure 13 indicates that the probability of the tag being positioned on another shelf or other floors is small. For example, the probability that a tag in A1 is incorrectly located to other floors is only 9%.

Our positioning method can provide a coarse-grained positioning accuracy for chemicals, and the accuracy we achieved was 92%. Although the positioning accuracy is not very high, this method can also reduce the search range for managers, reduce the workload and save time for them. Figure 14 shows the user interface of our system. The location information is the shelf number, the layer number, and the distance from the starting point in cm.

Although the RF-Detector basically meets our design requirements, it has yet to be improved in enhancing robustness. We require the reader to move along a straight line. However, it is impossible for the line to be strictly straight in practice. In future work, we should consider the deviation of the phase value caused by the irregular movement of the reader, and enhance the robustness of the RF-Detector.

## 7. Related Work

In recent years, the research on RFID technology has received more and more attention. RFID technology in warehouse management is mainly used to reduce the burden on managers and realize the automated management of the warehouse.

According to the previous description, there are two main problems to be solved in the chemical warehouse: One is detecting the remaining amount of chemicals, and the other is locating these chemicals. For the first problem, we can use indicators such as the reading frequency, phase value, and RSSI of the tag [14,15,16]. The phase difference and the received signal strength indication reflect the difference between the real received signal and the antenna transmitted signal [17,18,19]. As it is related to the environment, the phase value and RSSI of the tag are affected by the multipath effect in the environment [20,21]. Therefore, these parameters can be used to determine the remaining amount of chemicals. For positioning, traditional RFID-based indoor positioning methods include time of arrival (TOA), time difference of arrival (TDOA), angle of arrival (AOA), RSSI, etc. The TOA and TDOA algorithms require time synchronization and have high requirements for time measurement accuracy [22]. The AOA method mainly uses the angle of the RF wave from the reader antenna to the tag, so the angle information is extremely important. Angle information is usually obtained by an expensive professional antenna, but the cost of deploying a professional device is relatively high, which affects the feasibility of this method in actual deployment. The new measurement technology based on AOA for RFID positioning is a new technology that has been researched in recent years. The technology uses a custom RFID antenna-array to read the tag, and based on the inherent characteristics of the array, the angular information of the tag location can be obtained. PinPoint is a new AOA algorithm, which uses the rich information hidden in the RF signal in the interference environment to solve the positioning problem, and achieve the sub-meter positioning accuracy. Although the positioning accuracy has improved, it still does not meet our requirements.

LiBot [23] proposes a robot-assisted inventory management method, where the RFID system is deployed on the robot platform to realize automatic scanning of the shelves. Our work is inspired by the above works, and we raise new problems in the new application scenario and give detailed solutions. The experiments show that our system has high performance.

## 8. Conclusions

In this paper, we design the RF-Detector, which combines robot and RFID technology. We apply it to the management of chemical warehouses, where it solves two important issues: Detecting the remaining amount of chemicals and locating chemicals. For detection, we used two indicators: RSSI and read rate, and we use these two indicators as input to the SVM to determine whether the remaining amount of chemicals was sufficient. For localization, we constructed the phase profile of the RFID tag with the movement of the reader antenna, so we could get the position of the tags by curve fitting. The experimental results show that the RF-Detector achieved about 93% accuracy of the detection remaining amount and 92% positioning accuracy for chemicals. Therefore, the RF-Detector can be used in chemical warehouse to help managers detect the remaining amount of chemicals and locate those chemicals, greatly reduce their burden.

## Figures and Tables

**Figure 1 sensors-20-00123-f001:**
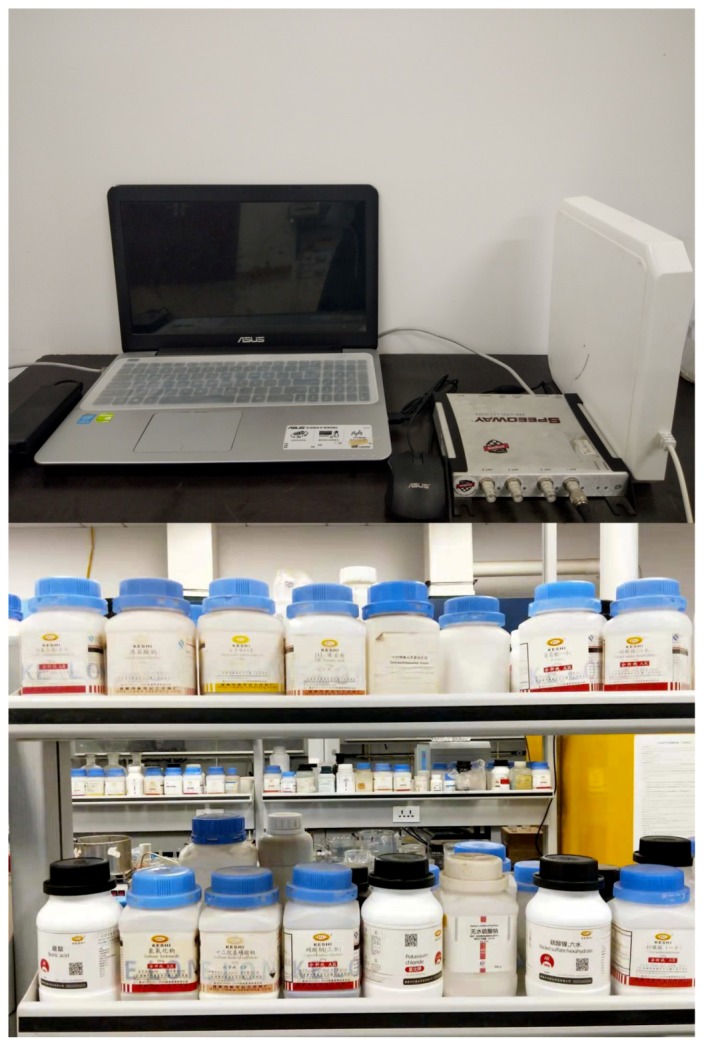
A corner of the chemical warehouse.

**Figure 2 sensors-20-00123-f002:**
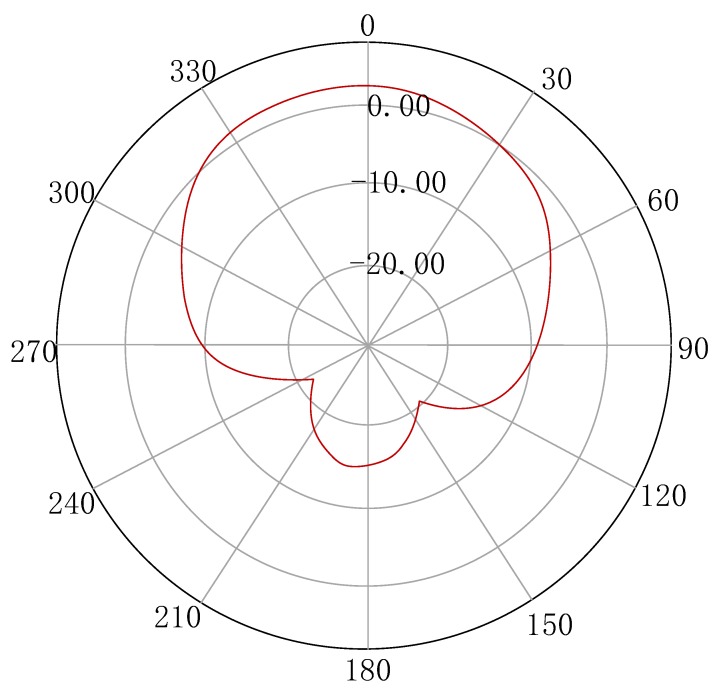
Radiation pattern of the ALR-8696-C antenna.

**Figure 3 sensors-20-00123-f003:**
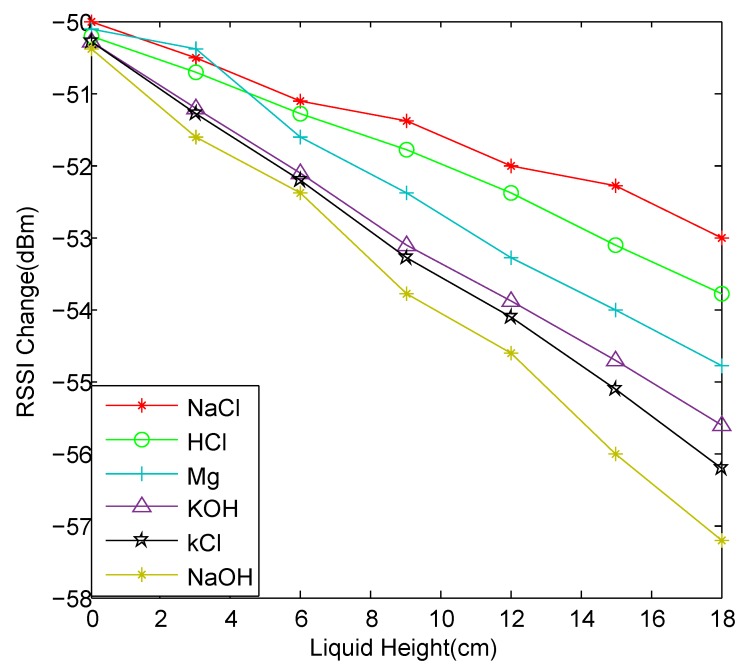
Received signal strength indication (RSSI) changes vs. liquid heights.

**Figure 4 sensors-20-00123-f004:**
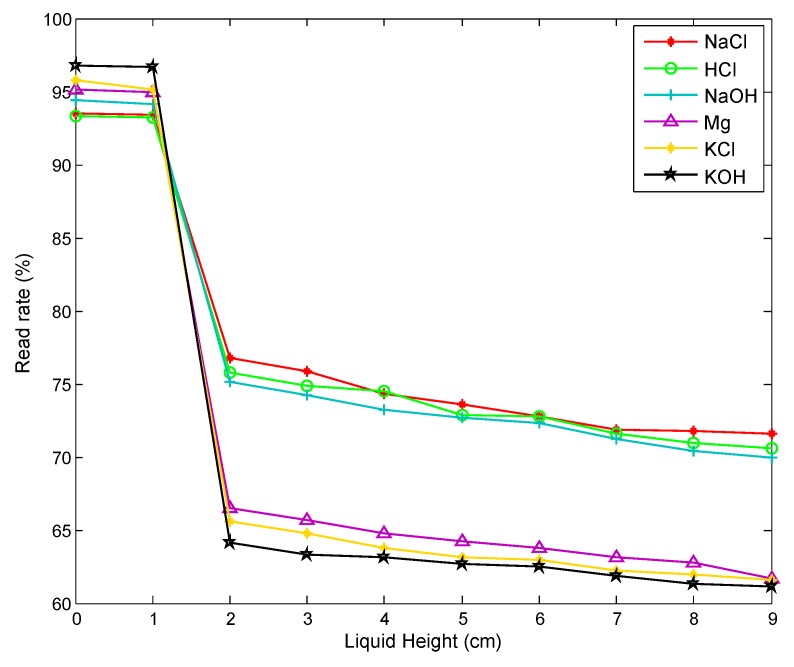
Read rate changes vs. liquid heights.

**Figure 5 sensors-20-00123-f005:**
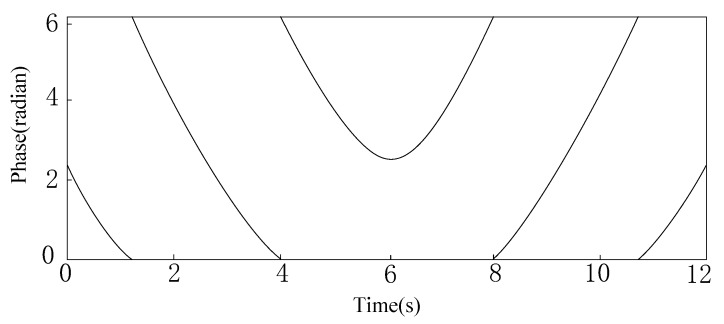
Theoretical phase profile.

**Figure 6 sensors-20-00123-f006:**
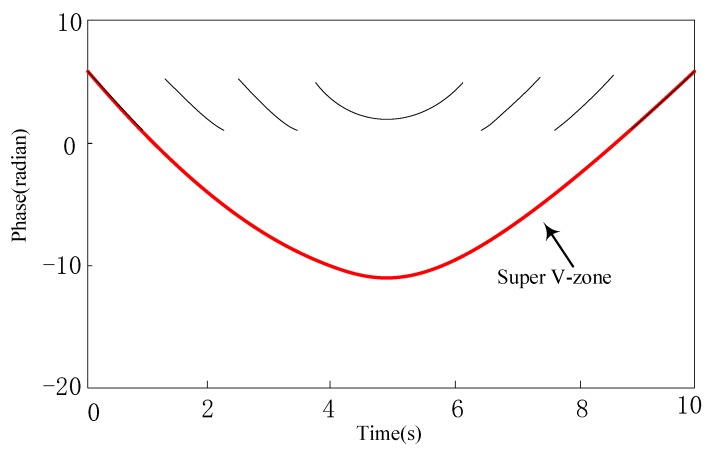
Super phase profile.

**Figure 7 sensors-20-00123-f007:**
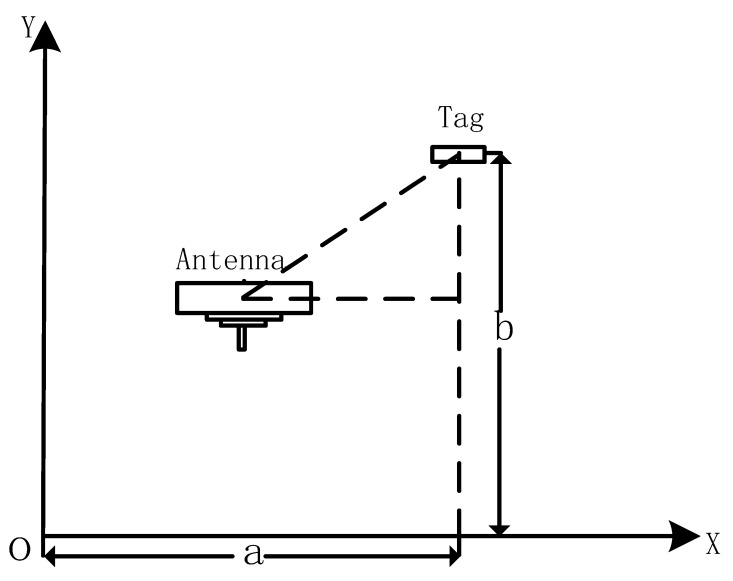
The distance between the antenna and the tag.

**Figure 8 sensors-20-00123-f008:**
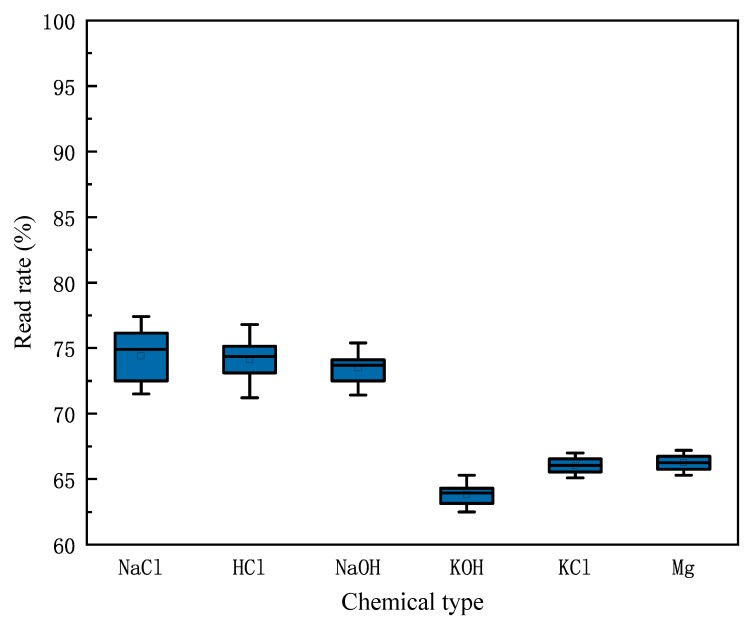
The read rate of chemical with sufficient margin.

**Figure 9 sensors-20-00123-f009:**
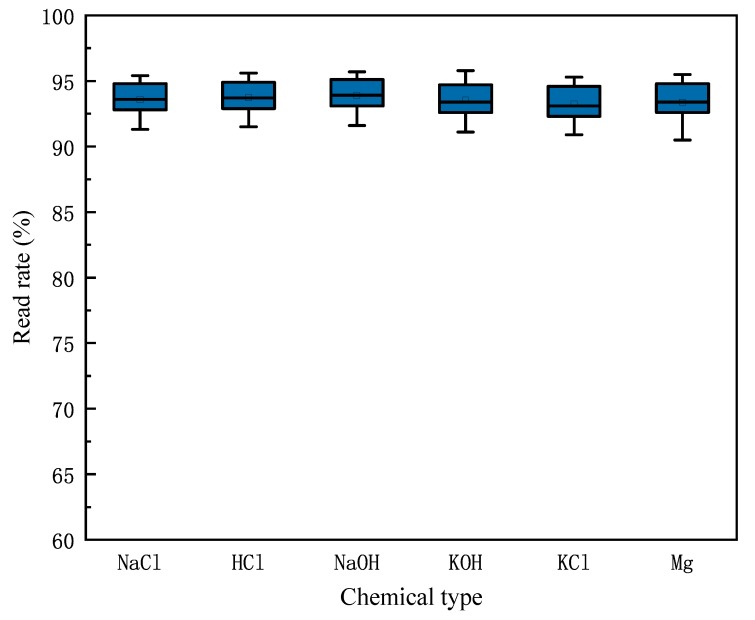
The read rate of chemical with insufficient margin.

**Figure 10 sensors-20-00123-f010:**
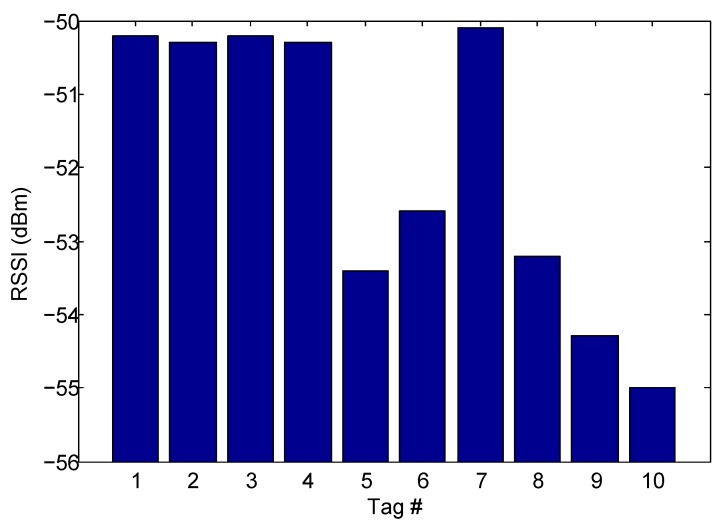
The RSSI of each tag.

**Figure 11 sensors-20-00123-f011:**
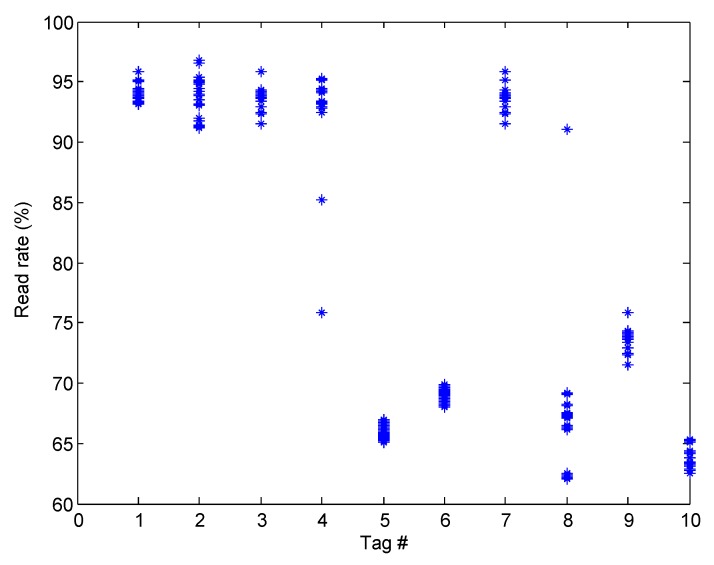
The read rate of each tag.

**Figure 12 sensors-20-00123-f012:**
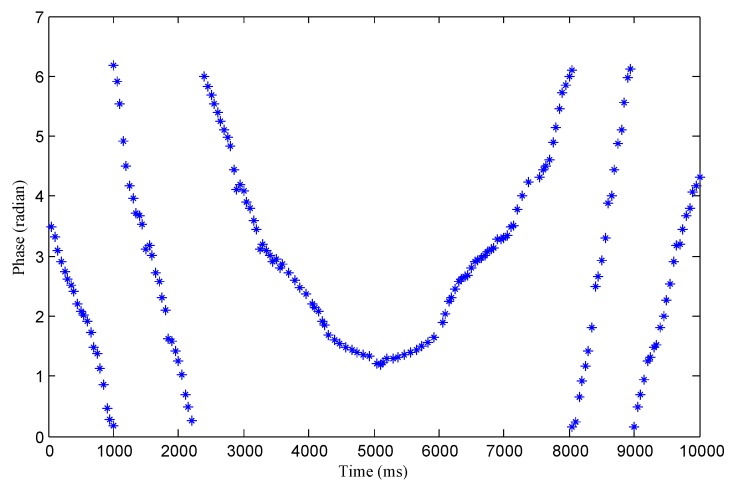
Measured phase profile.

**Figure 13 sensors-20-00123-f013:**
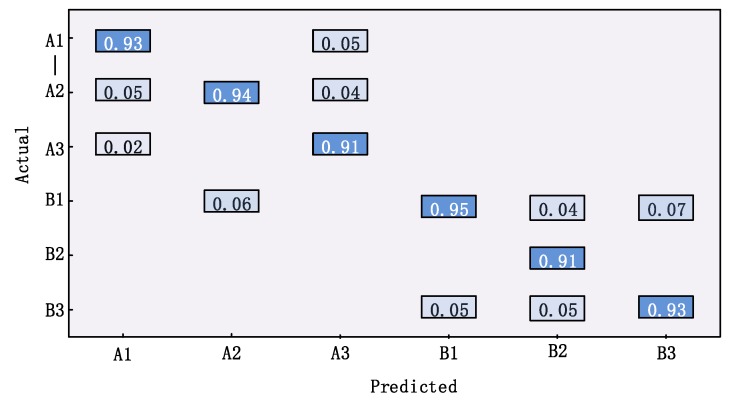
Localization accuracy.

**Figure 14 sensors-20-00123-f014:**
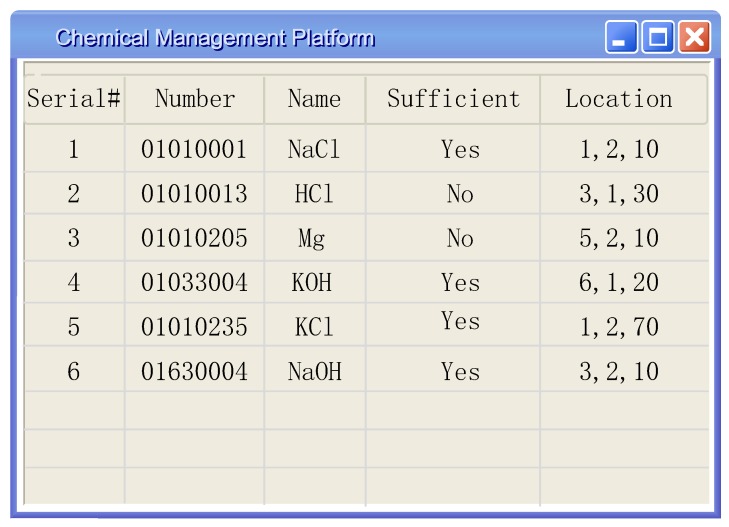
Chemical management interface.

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
