# Peer review of "Intelligent Management of Chemical Warehouses with RFID Systems"

_sensors, 2019, doi:10.3390/s20010123_

Round 1

Reviewer 1 Report

First, I want to congratulate you for your work. It is very practical and applied in a real scenario application. 

However, in my opinion you might review some aspects of your work:

1 - Regarding the level of chemical in each containers, you should not present your results in sufficient and insufficient levels. A better analysis would be to be specific about the level of chemical in each container. I would like to see a figure where you repeat the test for 1cm, 2cm, 3cm, etc of chemical in the bottle;

2 - I also would consider to improve the related work, you can be more exhaustive in the search for works that apply RFID in warehouses and industry, and reduce the introduction.

Reviewer 2 Report

The authors of this paper propose RF-Detector, which is a system combining a robot carrying an RFID reader to detect the level of stocked chemicals in a confined environment. The RSSI (Received Signal Strength Indicator) is considered along with the read rate to propose an algorithm to detect the remaining chemicals in containers.

The authors argue that by using RSSI with the read rate provides more accurate detections than when taking only into account RSSI. With this, the problem is seen as a simple classification to differentiate chemicals with enough remaining margin and those with insufficient margin. The most valuable contribution of this paper is the procedure for localization of chemicals which is based on a phase analysis and curve fitting. This is where the accuracy reported by the authors could be improved. Thus, at least a couple of remaining questions should be answered in this paper. The choice of an ImpinJ R420 reader and the directional antenna determines the physical limits of detection as well as the location of chemicals in the confined environment. Thus, can this performance be improved by distributing uniformly the location of the chemicals? What about the choice of RFID tags?

The paper is well structured; however, it must be fully revised for English grammar and typos. For example, starting with the abstract there is a sentence saying "Although some solutions used RFID technology ...", the word "used" must be surely replaced by "using". The Introduction section is labeled with "0" rather than with "1". In lines 38-39, the sentence "In this paper, we only use 39 robot technology to realize the movement of the reader." is confusing and badly written. This is the case of several parts of the paper; another example is the starting sentence of Section 5.

The justification of RF-Detector at the end of page 1 and at the start of page 2 should be better presented. References [7,8] do not accurately correspond to the case presented in this paper. Using robots with RFID readers has been used for a while in the last years, so more examples should be referenced in the paper to emphasize the contribution. Read for example the brief paper "Robot Employs RFID to Manage Warehouse Inventory" by Claire Swedberg in the RFID Journal. Examples like these are numerous in the literature, so it would be useful to add references like these in the paper.

Reviewer 3 Report

The paper presents an interesting work but it is confuse and the paper structure must be improved. Some technical details (radiation pattern of the antennas, the signal transmitted by THE RF-Detector, the measurement of the phase used for detection) are not given. The definition of the “read rate” is not clear. Concerning Figure 2, the text indicates: “as the liquid level rises, the RSSI of each chemical is also increasing”, but this figure shows the decrease of the RSSI! At the bottom of the page 6 one can read “If qi+1 – qi ≈ 2p, then it shows that there is a jump from 0 to 2p… In contrast, if qi+1 – qi ≈ 2p, then it shows that there is a jump from 2p to 0…” Where is the logic of this phrase? The structure of the paper is quite inappropriate: Figure 3 given in Section 3.1 shows the time variation of the phase, but the reader must wait till Section 3.3 to see the relation (6) which indicates how the phase is computed. The other problem is that (6) gives a unique value of the phase for a given value of the time but the Figure 3 shows for t = 1 s 3 different values of the phase! It is not clear how this result was obtained, why there are several curves on the figure.

In introduction, it is indicated that the detection accuracy can achieve 93%. How the rest of 7% cases are handled? Moreover, how the position accuracy of 92% was obtained?

At the end of the Introduction, one can read: “We next present the system architecture of RF-Detector in Section II” but just after this Introduction there is “1. System Architecture”. So, this section is II or 1 ? It is normal to use the same numbers indicated at the end of the introduction.

It is usual to have a space between the value of a parameter and its unit, as in page 3 row 84: “860-960 MHz”, not 13.56 MHz, as in page 3 row 85.

Page 3, row 113: write “where Pr…” because the phrase continues (it is not the beginning of a new phrase). The same modification is needed page 6, row159.

In Figure 8, the lowest RSSI value (-50 dBm) is placed on top, the largest value (-56 dBm) is placed on the bottom. Which is the logic of this quite unusual choice? Normally, the highest value must be places on the top, even if the RSSI values are negative. In this figure, the smallest rectangle corresponds to the largest value!

This paper needs a carefully revision.

Round 2

Reviewer 2 Report

The modifications made to the paper result in a good overall improvement. Thus, I have no more additional comments to make.

Author Response

Thank you for reviewing our paper.

Reviewer 3 Report

The paper presents an interesting work. The presentation is improved but the paper is still badly organized. Normally, the state of the art is presented in Introduction, not at page 14 (Section 7 Related Work).

For the phase measured for a tag, it is not clear how it is defined with a mathematical formula and how it is obtained. The notion of “detection accuracy” is used in Introduction (page 2, row 43) but its definition is given only in page 12 (rows 274-275). The definition of the “read rate”, used also in Introduction is given only in page 4 and it is still confusing. It is defined by “the total number of tag be actually read… divided by the maximum number of reads”. However, Figs. 8 and 9 suggest that for each tag a read rate can be obtained. Moreover, it is not clear how the “maximum number of reads” is chosen (imposed?). A good choice is to define each notion before it is used!

It is not clear how the lab manager has access to the date collected by the RF-Detector.

For Fig. 10, my suggestion was to place the RSSI value in increasing order, from -56 dBm up to -50 dBm and to use larger rectangles for larger RSSI values. For example, the largest rectangle must be placed for tag 1, because this tag has the largest RSSI value! The visual information does not match the RSSI value!

Some information concerning the RF-Detector is given in “Section 2 System Architecture”; some other information is given in “Section 5 Experiments”. In this Section 5 one can find the same information as in Section 3.1: “We prepare six identical glass bottles with a height of 18.5 cm and a diameter of 8.5 cm…”. This is quite perturbing for the reader. All the details concerning the measurement system must be given just once, in only one section.

Globally, the research work is useful and interesting, but badly structured, badly explained. Some typos must also be removed. However, this second version is improved comparing with the first one.
